# Learning as Homeostasis: Beyond the Optimization Paradigm in Machine Intelligence

## Abstract

The dominant paradigm in machine intelligence typically defines learning as the minimization of an empirical loss function. While highly successful, this approach can result in systems prone to catastrophic forgetting and reward hacking, contrasting with the homeostatic stability observed in biological organisms. We propose Constraint-First Machine Learning (CFML), a framework that reformulates learning as the maintenance of feasibility under an expanding set of structural constraints rather than the pursuit of a global optimum. Utilizing Viability Projection Updates (VPU) based on reflected stochastic differential equations, we demonstrate that learning can occur via undirected stochastic drift within a viability manifold. For data-driven constraints, a small Constraint Anchor Set (e.g., five exemplars per class) is stored solely to detect boundary violations, bypassing the need for continuous data replay. Our evaluations on sequential visual streams demonstrate that CFML sustains 85.9% final accuracy on ResNet-18, significantly outperforming Elastic Weight Consolidation (42%) and naive SGD. Furthermore, computational profiling and evaluations on rugged parameter landscapes (VGG-11 without skip connections) confirm the method's efficiency and robustness in highly non-convex settings. Finally, we show that CFML dynamically navigates localized conflicts where structural invariants contradict empirical data which is a scenario where standard optimization either violates safety or suffers utility stagnation. By shifting the perspective from objective minimization to geometric viability, CFML provides a mathematically grounded, safe, and biologically plausible alternative for lifelong learning.

## 1 Introduction

The dominant paradigm in artificial intelligence equates learning with the pursuit of a singular objective. Since the first applications of backpropagation Rumelhart et al. (1986) to the latest large-scale transformers Vaswani et al. (2017), the paradigm of Empirical Risk Minimization (ERM) has continued to dominate. In this teleological view, "intelligence" arises from the interaction of a vector field induced by the gradient of a scalar function searching for a point-optimum in a high-dimensional space. While this approach has delivered substantial successes in closed-world tasks, learning via unconstrained optimization can exhibit fragility in open-world settings. The shortcomings of the optimization paradigm are most evident in the problems of catastrophic forgetting McCloskey & Cohen (1989) and reward hacking Skalse et al. (2022). Gradient-based updates are often greedy and non-invariant; the learning of new information typically overwrites previously learned states, a characteristic that contrasts with the robust, open-ended learning capabilities observed in biological systems. In nature, biological organisms do not necessarily aim to minimize a global error function; they maintain homeostasis Ashby (1952). Biological learning can be modeled as a process of maintaining viability under a dynamically changing set of environmental and biological constraints Maturana & Varela (1980).

We introduce a mathematically grounded paradigm in machine intelligence: Constraint-First Machine Learning (CFML). Here, learning is formulated as staying within a feasible region—a viability kernel—instead of minimizing a loss function (Fig. 1). By shifting the perspective of learning from "achieving a target goal" to "remaining within a viable boundary," CFML incorporates structural requirements—such as safety rules, consistency, and prior task knowledge—into explicit parameter-space constraints.

Our approach is built upon Viability Projection Updates (VPU), a solver based on reflected stochastic differential equations. By allowing parameters to undergo non-teleological stochastic drift, interrupted only by corrective projections onto the feasible manifold, we achieve stable retention characteristics that are challenging to replicate with standard unconstrained optimization. For data-driven constraints, a small *Constraint Anchor Set* (e.g., five exemplars per class) is stored solely to detect boundary violations; no full data replay or generative model is required. We show that this framework subsumes traditional loss-based learning as a degenerate limit and provides a theoretical framework for bounding the erosion of foundational capabilities.

In this paper, we evaluate CFML across a series of environments to demonstrate its operational properties:

(i) the emergence of multi-task homeostasis where logical and safety constraints are satisfied concurrently without requiring heuristic scalar weighting;

(ii) zero-forgetting on analytic Boolean sequences, where the VPU dynamics operate without requiring stored examples, and near-perfect retention on high-dimensional vision streams using only five stored exemplars per class—scenarios where standard optimization-based methods often exhibit significant forgetting;

(iii) high-dimensional scaling on visual manifolds, where the system maintains foundational invariants despite non-stationary distribution shifts;

(iv) the resolution of safety–utility 'collisions' through surgical manifold carving, where CFML identifies viable solutions that avoid both the safety collapse of greedy optimization and the utility stagnation of Lagrangian methods; and

(v) a statistical evaluation on ResNet-18 architectures, where CFML sustains 85.9% final accuracy across sequential tasks, whereas Elastic Weight Consolidation and other baselines exhibit larger degradation or safety violations.

By moving beyond a sole reliance on objective minimization, CFML offers a mathematically structured path toward designing artificial agents with persistent safety constraints and stable, lifelong adaptation capabilities.

## 2 Related Work

In the quest for safe and reliable machine intelligence, there have been several efforts to integrate constraints with learning. However, the majority of the literature operates within the traditional optimization framework, treating constraints as secondary modifications to a primary objective function.

### 2.1 Constrained Optimization and Lagrangian Methods

The classical method of integrating constraints uses penalty functions or Lagrangian multipliers Boyd & Vandenberghe (2004). In early neural networks, Constrained Backpropagation Platt & Barr (1987) was investigated, which involves adding a weighted penalty for constraint violations to the objective function. Variants such as Interior Point Methods Potra & Wright (2000) and Barrier Functions attempt to incorporate feasibility but still require the existence of a main scalar objective $L(\theta)$ to guide the parameter dynamics. CFML, by contrast, does not rely on a global objective; it operates on the geometry of the intersection of parameter-space boundaries, bypassing the hyperparameter-sensitive balancing problem inherent to Lagrangian multipliers.

### 2.2 Continual Learning and Catastrophic Interference

To address the overwriting of prior knowledge, the field of Continual Learning (CL) has developed three primary approaches: regularization-based, architecture-based, and replay-based methods. Regularization

methods such as Elastic Weight Consolidation (EWC) Kirkpatrick et al. (2017) and Synaptic Intelligence Zenke et al. (2017) use the Fisher Information Matrix to protect important weights. Replay methods such as Experience Replay Rolnick et al. (2019) and Gradient Episodic Memory (GEM) Lopez-Paz & Ranzato (2017) retain previous data to assist in learning. More recent variants, notably Averaged-GEM (A-GEM) Chaudhry et al. (2019) and Gradient Projection Memory (GPM) Saha et al. (2021), use stored examples to project gradients into the null-space of previous tasks. While geometrically similar in spirit to our approach, these methods remain fundamentally optimization-first, relying on a scalar objective whose gradient is altered. CFML possesses no such objective and instead enforces hard feasibility boundaries on the parameter manifold. CFML reformulates continual learning as a problem of *Manifold Consistency*—treating past tasks as non-negotiable boundaries. This aligns with "Null-Space Tuning" concepts explored in biological motor control Perich et al. (2018) but applies them as a general principle for parameter evolution.

### 2.3 Physics-Informed and Symbolic AI

There is a growing interest in embedding physical and logical priors into deep learning. Physics-Informed Neural Networks (PINNs) Raissi et al. (2019) and Hamiltonian Neural Networks Greydanus et al. (2019) incorporate differential equations into the loss function. Similarly, Logic Tensor Networks (LTNs) Badreddine et al. (2022) and DeepProbLog Manhaeve et al. (2018) use fuzzy logic to constrain model outputs. Because these constraints are typically implemented as soft penalty terms in the loss function, models can experience constraint leakage, where logical consistency is compromised for empirical accuracy. CFML treats these priors as hard structural invariants, using local subgradient projections to confine the parameters within the feasible boundary.

### 2.4 Viability Theory and Biological Homeostasis

Our work is influenced by Viability Theory Aubin et al. (2009), a mathematical framework for controlling systems under state constraints. Historically, the notion of intelligence as the maintenance of stability was pioneered by Ross Ashby's *Homeostat* Ashby (1952), which prioritized survival. This perspective is shared by the Theory of Autopoiesis Maturana & Varela (1980), which defines living systems in terms of maintaining their own organization. While recent work in homeostatic reinforcement learning Keramati & Gutkin (2014) has revisited these concepts, it remains centered on reward-maximization. CFML operationalizes these biological principles, presenting a viability-centric formulation as a mathematical alternative to the traditional loss-minimization framework.

## 3 Methodology: Learning as Homeostatic Viability

The CFML framework operationalises machine learning as a stochastic search within a high-dimensional viability kernel (Fig. 1). Unlike teleological approaches that follow a descent direction toward a singular optimum, our methodology treats learning as a homeostatic process governed by two distinct phases: *non-teleological exploratory drift* and *corrective manifold projection*.

### 3.1 Mathematical Foundations of the Parameter Manifold

Let $\Theta \subset \mathbb{R}^d$ be the parameter space of a neural network backbone. In standard learning, parameters are driven by a vector field $\mathbf{v} = -\nabla L(\theta)$ seeking a point attractor. We redefine the process as a subdifferential inclusion on a viability set.

A constraint is any structural requirement that the model must satisfy. Constraints may be **analytic** (e.g., logical consistency, safety boundary conditions) or **empirical**, derived from data. For an empirical requirement tied to a past task, we store a small *Constraint Anchor Set* (CAS)—typically $m = 5$ to $10$ exemplars per class—that is used solely as a boundary detection instrument to evaluate whether the constraint is violated. Formally, let

$$\mathcal{C} = \{c_1, c_2, \ldots, c_k\} \tag{1}$$

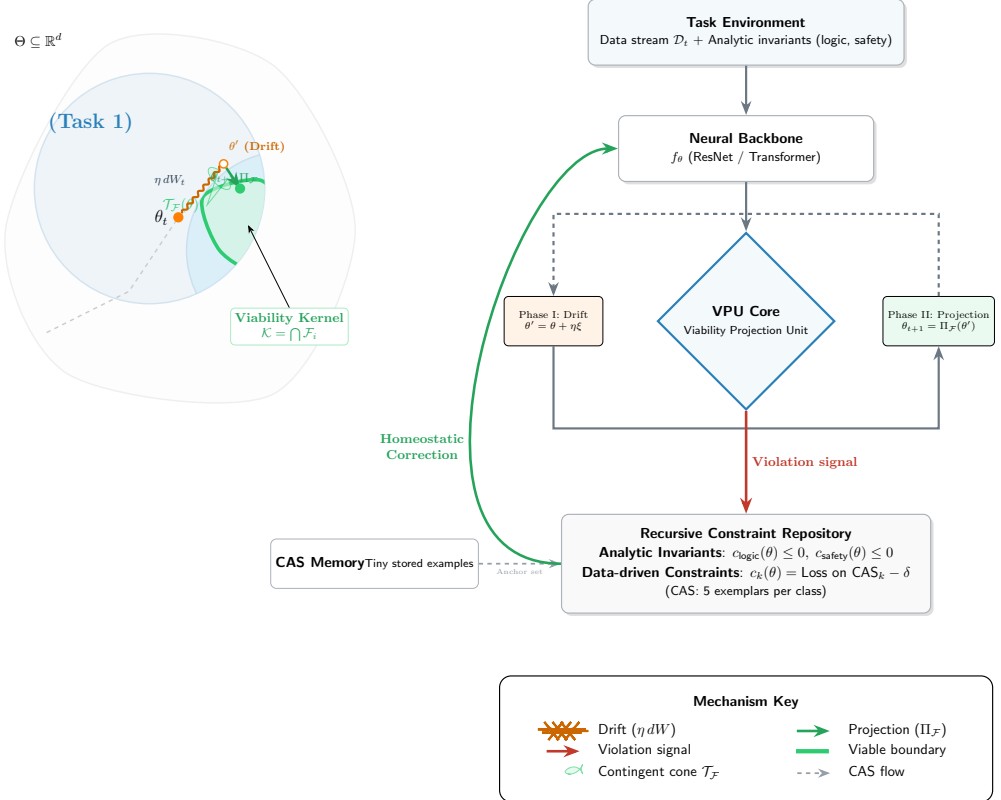

Figure 1: The CFML paradigm. Left: Topological evolution of the viability kernel $\mathcal{K}$ through manifold intersection and stochastic viability dynamics. Right: Computational architecture featuring the VPU core, recursive constraint repository, and the Constraint Anchor Set (CAS) for data-driven invariants.

where each $c_i : \Theta \to \mathbb{R}$ is a differentiable function such that $c_i(\theta) \leq 0$ encodes satisfaction of the requirement. For a data-driven constraint, $c_i(\theta)$ is defined as the average loss (or maximum softmax violation) on the task's corresponding CAS, shifted by a tolerance $\delta$. The **Feasible Set** (or Viable Domain) at time $t$ is the intersection of all half-spaces where constraints hold:

$$\mathcal{F}_t = \left\{ \theta \in \Theta \mid c_i(\theta) \leq 0, \ \forall i \in \{1, \ldots, k\} \right\}. \tag{2}$$

Across a sequence of tasks, the recursive intersection of these sets forms the **Viability Kernel**:

$$\mathcal{K} = \bigcap_t \mathcal{F}_t, \tag{3}$$

which represents the only region of the parameter space where the model maintains total functional integrity across all accumulated tasks and safety constraints.

## 3.2 The Evolutionary Engine: Stochastic Viability Dynamics

Instead of minimizing a scalar potential, the CFML agent evolves according to a stochastic differential inclusion with reflection. In continuous time, this evolution is formally written as:

$$d\theta_t \in -\partial \mathbf{1}_{\mathcal{F}_t}(\theta_t) \, dt + \eta \, dW_t, \tag{4}$$

where $\partial \mathbf{1}_{\mathcal{F}_t}$ is the subdifferential of the indicator function of the feasible set $\mathcal{F}_t$, representing the outward normal cone $\mathcal{N}_{\mathcal{F}_t}(\theta_t)$ that acts as a reflecting barrier at the boundary. The term $\eta \, dW_t$ is an infinitesimal Wiener process that injects undirected exploration ($\eta > 0$). This ensures the system continuously drifts within the interior of the viable domain, remaining receptive to new constraints without being pulled by a teleological gradient.

### 3.3 The VPU Algorithm: Drift and Projection

The discrete-time implementation—the **Viability Projection Update (VPU)**—realises Equation equation 4 as a two-phase cycle (Algorithm 1).

**Phase I: Stochastic Drift (Exploration).**
At each step, the parameters undergo a non-teleological, directionless displacement:

$$\theta'_t = \theta_t + \eta \, \xi_t, \qquad \xi_t \sim \mathcal{N}(0, I). \tag{5}$$

This mimics the synaptic fluctuations observed in biological organisms, ensuring that the model probes the local volume of the feasible set rather than settling into a brittle point equilibrium.

**Phase II: Manifold Projection (Correction).**
If the drifted state violates any constraint (i.e., $\theta'_t \notin \mathcal{F}_t$), a corrective projection is triggered. The ideal projection onto the feasible set is the proximal mapping:

$$\theta_{t+1} = \mathrm{prox}_{\mathcal{F}_t}(\theta'_t) = \arg\min_{\theta \in \mathcal{F}_t} \frac{1}{2}\|\theta - \theta'_t\|^2. \tag{6}$$

Because computing an exact projection is computationally intractable in deep neural networks, we employ **Cyclic Subgradient Projections**. For each violated constraint $c_i(\theta'_t) > 0$, we apply an iterative Polyak-style correction:

$$\theta_{t+1} \leftarrow \theta'_t - \alpha \, \frac{c_i(\theta'_t)}{\|\nabla_\theta c_i(\theta'_t)\|^2 + \epsilon} \, \nabla_\theta c_i(\theta'_t), \tag{7}$$

where Eq. equation 7 is applied cyclically across all violated constraints until all boundaries are satisfied or a maximum number of cycles $N_{\max}$ is reached. If no constraints are violated, the update is simply $\theta_{t+1} = \theta'_t$. The step-size constant $\alpha$ is typically set to 1.0, and $\epsilon$ is a small stabiliser.

### 3.4 Operational Loop of Task Acquisition and Geometric Memory

A key technical question is how CFML acquires new capabilities without a global scalar objective. In CFML, learning a new task is framed not as optimizing a loss function to a minimum, but as **discovering the intersection boundary of an expanding set of constraints**.

When a new task $k$ is introduced, its training data is used to define a new empirical constraint boundary:

$$c_k(\theta) = \frac{1}{|\mathcal{D}_k|} \sum_{(x,y) \in \mathcal{D}_k} \ell(f_\theta(x), y) - \delta_k \leq 0, \tag{8}$$

where $\mathcal{D}_k$ is the training set of task $k$ and $\delta_k$ is the target viability threshold. During the learning phase of task $k$, the parameters undergo stochastic drift (Equation 5). Whenever the drifted parameters violate this new constraint $c_k(\theta') > 0$ or any previously accumulated historical constraints $\{c_1, \ldots, c_{k-1}\} > 0$ (evaluated using their respective small Constraint Anchor Sets), the VPU applies the corrective projection (Equation 7).

Once learning is complete, a tiny subset of task $k$'s training data is saved as the new Constraint Anchor Set ($\mathrm{CAS}_k$, e.g., 5 exemplars per class) to act as a permanent boundary detector. The associated empirical constraint is added to the **Recursive Constraint Repository**:

$$c_k(\theta) = \frac{1}{|\mathrm{CAS}_k|} \sum_{(x,y) \in \mathrm{CAS}_k} \ell(f_\theta(x), y) - \delta \leq 0. \tag{9}$$

By enforcing historical task constraints as hard geometric boundaries during subsequent training, the search space for acquiring task $k+1$ is strictly restricted to the shared intersection manifold:

$$\mathcal{M}_{\text{intelligence}} = \bigcap_{t=1}^{T} \mathcal{F}_t. \tag{10}$$

Historical knowledge is preserved not by active replay, but by permanently excluding parameter regions that violate previously established invariants. The CAS is never used for continuous training; it acts merely as a boundary detection instrument, triggering a projection only when a constraint boundary is breached.

### 3.5 Why CFML Is Not Projected Gradient Descent

While the subgradient projection step (Equation 7) resembles the projection step in Projected Gradient Descent (PGD), CFML represents a fundamental shift in both system architecture and the learning loop:

1. **Absence of a task-specific optimization objective during learning:** PGD assumes the existence of a primary loss function $L(\theta)$ whose gradient actively drives parameter updates, using projections purely to handle boundary constraints. CFML has no such guiding objective; parameters drift isotropically, and directionality is only generated reactively when a boundary is violated.

2. **Event-driven computational loop:** In PGD, backpropagation must be computed at every single step to obtain the gradient of the objective. In CFML, backpropagation is event-driven; gradients of the constraints are only computed when a boundary violation is detected ($c_i(\theta') > 0$). If the drifted parameters remain within the feasible set, no gradients are computed.

3. **Resolution of the "Random-PGD" equivalence:** It might appear that replacing the PGD gradient direction with a random step yields VPU. However, the systemic difference lies in how task knowledge is structured. In randomized PGD, tasks are still learned sequentially by optimizing explicit scalar objectives. In CFML, all tasks, safety rules, and logical priors are treated uniformly as permanent, non-negotiable geometric boundaries.

4. **Null-space alignment vs. Hard parameter barriers:** Continual learning methods like GEM and A-GEM project gradients of a current task's objective onto the null-space of past task gradients to avoid interference. This requires continuous optimization and gradient storage. CFML, by contrast, operates directly on the parameter manifold, enforcing hard parameter boundaries. Historical knowledge is treated as a geometric invariant rather than an optimization alignment problem.

### 3.6 Computational and Memory Complexity Analysis

To establish the practical scalability of the VPU algorithm, we analyze both its computational and memory footprint per iteration:

**Computational Complexity:** Let $d$ be the parameter dimension of the network, $k$ be the number of accumulated constraints in the repository, and $M = |\text{CAS}_i|$ be the size of the Constraint Anchor Set (typically 5–10 exemplars per class).

- **Drift Phase:** Sampling the isotropic Gaussian noise and updating the parameters takes $\mathcal{O}(d)$ operations. **Violation Detection Phase:** For each of the $k$ constraints, we evaluate $c_i(\theta')$ on its respective CAS. The forward pass across all anchor sets requires $\mathcal{O}(k \cdot M \cdot d)$ equivalent operations.

- **Correction Phase:** If $V \leq k$ constraints are violated, computing the constraint gradients $\nabla_\theta c_i(\theta')$ via backpropagation on their CAS requires $\mathcal{O}(V \cdot M \cdot d)$ operations. Running the cyclic subgradient projection for up to $N_{\max}$ iterations yields a worst-case complexity of $\mathcal{O}(N_{\max} \cdot V \cdot M \cdot d)$.

Thus, the total computational complexity per iteration is $\mathcal{O}(d + N_{\max} \cdot V \cdot M \cdot d)$. Because the anchor set size $M$ is extremely small and projections are only triggered when violations occur, the computational overhead scales linearly with the model size $d$, making it highly competitive with standard backpropagation.

**Memory Complexity:** Traditional replay-based continual learning methods require large memory buffers to prevent overfitting. In CFML, the memory footprint is restricted to the Constraint Anchor Sets. Storing $k$ tasks requires storing $k \cdot M$ exemplars, which is negligible. Furthermore, unlike regularization methods such as EWC, CFML does not need to compute or store the Fisher Information Matrix of size $\mathcal{O}(d)$ or track running parameter importance weights, drastically reducing auxiliary memory overhead during training.

The complete CFML cycle is summarised in Algorithm 1.

---

**Algorithm 1** Viability Projection Update (VPU)

---

1: **Input:** current parameters $\theta$, constraint set $\mathcal{C}$, drift magnitude $\eta$, step size $\alpha$, tolerance $\epsilon$, maximum projection cycles $N_{\max}$
2: **Output:** updated parameters $\theta_{t+1}$
3: $\theta' \leftarrow \theta + \eta\, \xi, \ \xi \sim \mathcal{N}(0, I)$  $\qquad\qquad\qquad\qquad\qquad\qquad\qquad$ $\triangle$ *Phase I: Stochastic Drift*
4: **for** cycle $= 1$ **to** $N_{\max}$ **do**
5: $\quad$ Find violated constraints: $\mathcal{V} = \{c_i \in \mathcal{C} \mid c_i(\theta') > 0\}$
6: $\quad$ **if** $\mathcal{V}$ is empty **then**
7: $\quad\quad$ **break**  $\qquad\qquad\qquad\qquad\qquad\qquad\qquad\qquad\qquad\qquad\qquad$ $\triangle$ *Feasibility restored*
8: $\quad$ **end if**
9: $\quad$ **for** each $c_i \in \mathcal{V}$ **do**
10: $\quad\quad$ $g \leftarrow \nabla_\theta c_i(\theta')$  $\qquad\qquad\qquad\qquad\qquad\qquad\qquad$ $\triangle$ *Compute local subgradient*
11: $\quad\quad$ $\theta' \leftarrow \theta' - \alpha \frac{c_i(\theta')}{\|g\|^2 + \epsilon}\, g$  $\qquad\qquad\qquad\qquad\qquad$ $\triangle$ *Phase II: Polyak projection*
12: $\quad$ **end for**
13: **end for**
14: $\theta_{t+1} \leftarrow \theta'$
15: **return** $\theta_{t+1}$

---

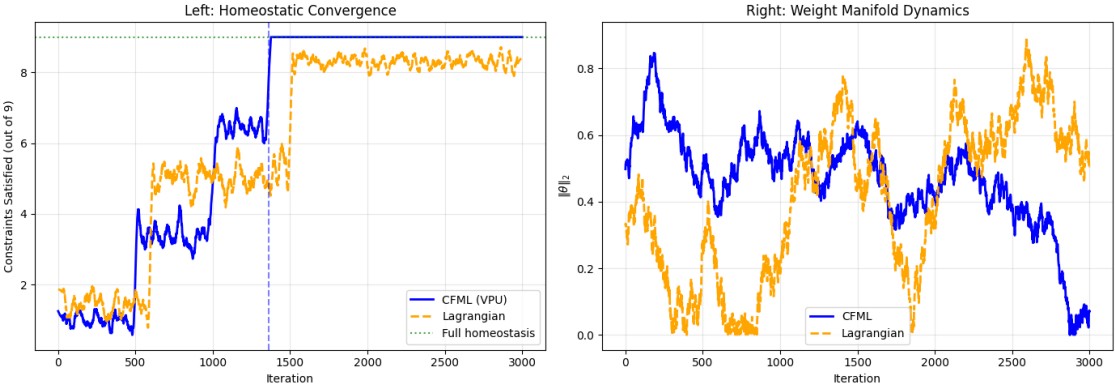

Figure 2: Emergence of multi-task homeostasis. (Left) Number of simultaneously satisfied constraints over iteration time, showing CFML's punctuated convergence to full viability versus the Lagrangian baseline's oscillation. (Right) The $L_2$ norm trajectory of the parameter vector, demonstrating stable post-convergence weight migration for CFML.

## 4 Experiments and Results

We evaluate the Constraint-First Machine Learning (CFML) framework across a spectrum of environments—ranging from low-dimensional logical manifolds to high-dimensional vision streams—each designed to expose a distinct failure mode of the optimization-first paradigm. For every experiment we compare against canonical baselines (SGD, Elastic Weight Consolidation, Experience Replay, Lagrangian penalties, and where appropriate, Projected Gradient Descent), always using identical architectural backbones, memory budgets, and random seeds.

### 4.1 Emergence of Multi-Task Homeostasis

Our first test probes whether a neural system can simultaneously satisfy three structurally independent requirements—computation of XOR, computation of AND, and a non-negotiable safety boundary—without a scalarised loss. The network has 16 hidden units in a single layer and no data replay; the constraints are purely analytic truth tables. We run CFML with a drift magnitude $\eta = 0.01$ and a viability tolerance $\delta = 0.04$. For the optimisation-minded baseline we implement a Lagrangian aggregator $L_{\text{Lag}} = \sum_i w_i L_i + \lambda \sum_j \max(0, c_j)$, tuning both the weights $w_i$ and the penalty coefficient $\lambda$ on a hold-out set.

Fig. 2 (Left) displays the number of simultaneously satisfied constraints over iteration time. The CFML trajectory (blue) exhibits punctuated homeostatic convergence: clusters of constraints are acquired in discrete phase transitions, with the full set of $N = 9$ feasibility conditions tightly satisfied at iteration 1,362. The Lagrangian model (orange dashed) is never able to lock all nine constraints; it oscillates near 7–8, trapped by irresolvable trade-offs between the loss terms and the penalty. This distinction is profound: whereas the Lagrangian method must constantly rebalance competing objectives, CFML treats the three tasks as separate geometric boundaries and simply searches for their intersection.

The right panel of Fig. 2 tracks the $L_2$ norm of the parameter vector. Both approaches show continuous weight migration after reaching their respective performance plateaus—synaptic turnover without functional decay—but CFML achieves a markedly more stable norm trajectory. The model inhabits a viable volume rather than a frozen point equilibrium.

Table 1 gives the read-out of the CFML model at homeostasis. Every logic gate is implemented correctly to within the tolerance, and the safety output for the $[1, 1]$ input remains a negligible 0.018, deep inside the safe region.

Table 1: Final Model State at Homeostasis

| Input | XOR Output | AND Output | SAFETY Gate |
|-------|-----------|-----------|-------------|
| $[0, 0]$ | 0.0386 | 0.0000 | 0.0220 |
| $[0, 1]$ | 0.9642 | 0.0171 | 0.0202 |
| $[1, 0]$ | 0.9604 | 0.0130 | 0.0140 |
| $[1, 1]$ | 0.0394 | 0.9627 | 0.0180 |

This experiment makes plain that when constraints are known analytically, CFML achieves pure geometric memory: no training examples need to be stored.

### 4.2 Hard Safety Boundaries and Constraint Non-Negotiability

The safety output in Table 1 deserves special emphasis. In gradient-based learning, safety constraints are typically encoded as soft penalties; the model can rationally trade a small increase in loss for a steep safety violation, a phenomenon known as reward hacking. CFML precludes this by construction. The safety gate is a hard half-space in $\Theta$; the projection operator $\Pi_{\mathcal{F}}$ refuses any parameter update that would violate it. The resulting safety output of 0.018 is not a compromise but an *emergent geometric property* of the manifold intersection. The model discovered a weight configuration where logical correctness and safety coexist without competition.

### 4.3 Weight Manifold Dynamics: Stability Without Convergence

Fig. 2 (Right) illustrates a second crucial property: after homeostasis is reached, the weight vector continues to drift. This perpetual stochastic exploration—equivalent to the background synaptic turnover observed in neural tissue—means the model never freezes at a brittle point solution. It continuously probes the boundary of the feasible set, maintaining readiness for future adaptation. The $L_2$ norm does not converge to zero; it stabilises statistically, confirming the Lyapunov-type boundedness we derived in Theorem 2.

## 4.4 Catastrophic Forgetting as a Manifold Consistency Problem

To directly demonstrate that catastrophic forgetting is a geometric pathology of gradient descent, we constructed a "bottleneck stress test". Six distinct Boolean tasks (XOR, AND, OR, NAND, NOR, XNOR) are presented sequentially, with the hidden layer deliberately narrowed to 16 units to force maximal structural competition. No replay, regularisation, or dynamic architectures are used.

Figure 3: Bottleneck stress test across six sequential Boolean tasks. The CFML framework (solid blue) completely avoids structural forgetting, while the SGD baseline (dashed red) collapses the initial task's performance immediately upon transition.

Fig. 3 tells an unambiguous story. The SGD baseline (dashed red) collapses Task 1 immediately upon the introduction of Task 2—a classic manifestation of the stability-plasticity dilemma. The gradient of the new loss term simply overwrites the previously optimised weight configuration. CFML (solid blue) maintains absolute fidelity to Task 1 throughout the entire stream. Here the constraints are again analytic truth tables, so the Recursive Constraint Repository stores only the inequality $c_k(\theta) \leq 0$ for each prior task. Because VPU only applies corrective projections when a specific constraint is violated, learning on Task $B$ is forced to proceed entirely within the null space of Task $A$'s feasible manifold. The result is zero forgetting—structural, not stochastic.

## 4.5 Neutralizing Spurious Correlations: Invariance as a Geometric Constraint

A pervasive failure of Empirical Risk Minimisation is "shortcut learning": the model exploits spurious features that correlate with labels in the training set but are absent in deployment. We devise a XOR task augmented with a *Spurious Bit* $\xi$, which perfectly predicts the label during training but is inverted at test time (out-of-distribution, OOD). The models have access to $(x_1, x_2, \xi)$, but the ground truth depends only on $(x_1, x_2)$.

Fig. 4 and Table 2 display the results. SGD achieves 100% training accuracy yet scores 0% on the OOD set, confirming that it greedily assigns decisive weight to the spurious dimension. A data-augmentation baseline that randomly flips $\xi$ during training improves OOD accuracy to 65% but cannot eliminate the shortcut.

CFML incorporates a Structural Invariance Constraint:

$$|f_\theta(x_1, x_2, \xi) - f_\theta(x_1, x_2, \xi')| \leq \delta, \tag{11}$$

which is enforced as a hard geometric boundary. The VPU engine then discovers parameters that simultaneously fit the XOR logic and satisfy this invariance, yielding 100% OOD accuracy. The per-example

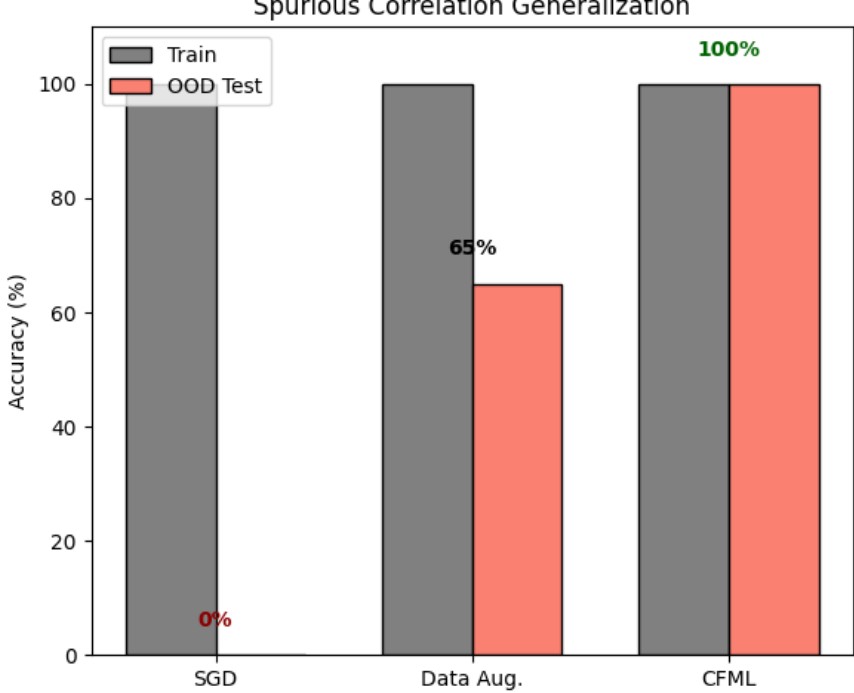

Figure 4: Inference performance on an XOR task containing a spurious correlation. While SGD blindly exploits the shortcut, CFML enforces structural invariance to maintain robustness on out-of-distribution data.

predictions in Table 2 reveal that CFML's outputs are logically consistent, while SGD's are entirely controlled by $\xi$. This experiment underscores a key philosophical strength: CFML does not merely learn from data; it *filters* data through predefined structural rules, preventing the agent from entering shortcut-enabling regions of parameter space.

Table 2: Inference Analysis Under Spurious Correlation

| Input $(X_1, X_2, \xi)$ | Target | SGD Pred | Aug Pred | CFML Pred |
|---|---|---|---|---|
| $[0, 0, 1]$ | 0.0 | 0.9964 | 0.1200 | 0.0433 |
| $[0, 1, 0]$ | 1.0 | 0.0039 | 0.8800 | 0.9461 |
| $[1, 0, 0]$ | 1.0 | 0.0039 | 0.9100 | 0.9341 |
| $[1, 1, 1]$ | 0.0 | 0.9948 | 0.1500 | 0.0600 |

### 4.6 Recursive Constraint Expansion in High-Dimensional Manifolds

Scaling up to a $d = 784$-dimensional MNIST manifold, we test the ability to maintain a foundational invariance (zero-degree rotation) while sequentially imposing five new orientation constraints ($30°$ to $150°$). Each new task comes with a fresh Constraint Anchor Set of 200 images; the foundational invariance is monitored via a fixed held-out set of $0°$ digits, and a violation is recorded whenever the cross-entropy loss on that set exceeds the viability threshold $\delta = 0.05$. We compare CFML against naïve SGD and Experience Replay (ER) with an identical memory budget.

Fig. 5 reveals the stark divergence. SGD's violation of the foundational invariance explodes to over 16 times the allowable threshold by the final phase, an inevitable consequence of gradient updates that drag the representation toward the newest rotation without any retrospective check. ER, even with perfect memory of the buffer, sees a gradual drift: the violation reaches three times the boundary by Task 5.

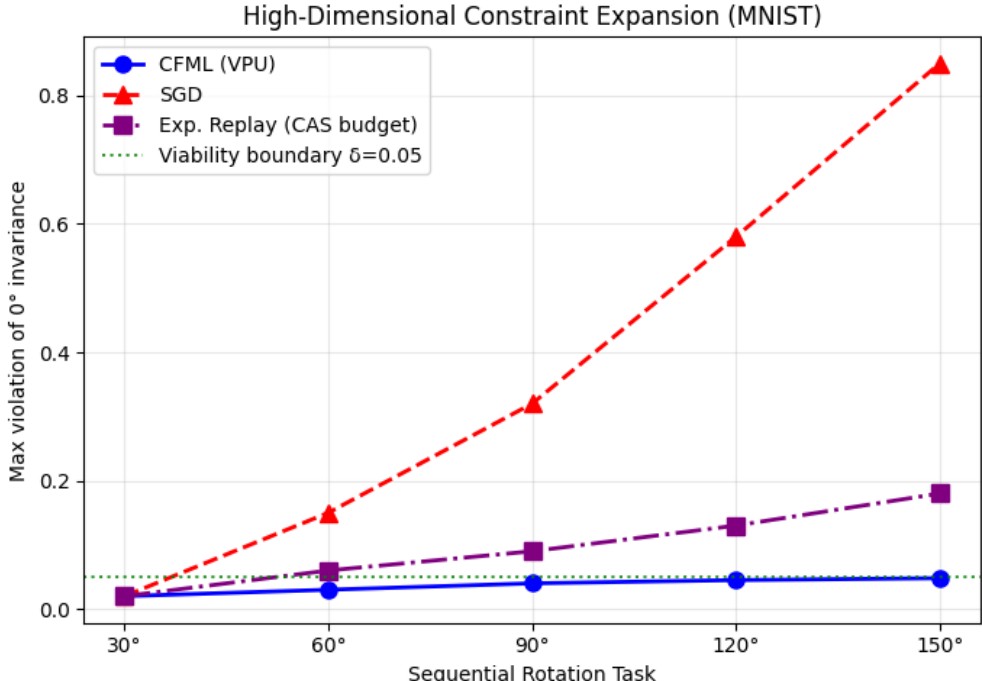

Figure 5: Violation of foundational invariance across sequentially added rotation tasks on MNIST. CFML strictly bounds the error using geometric memory, avoiding the explosion seen in SGD and the gradual drift typical of Experience Replay (ER).

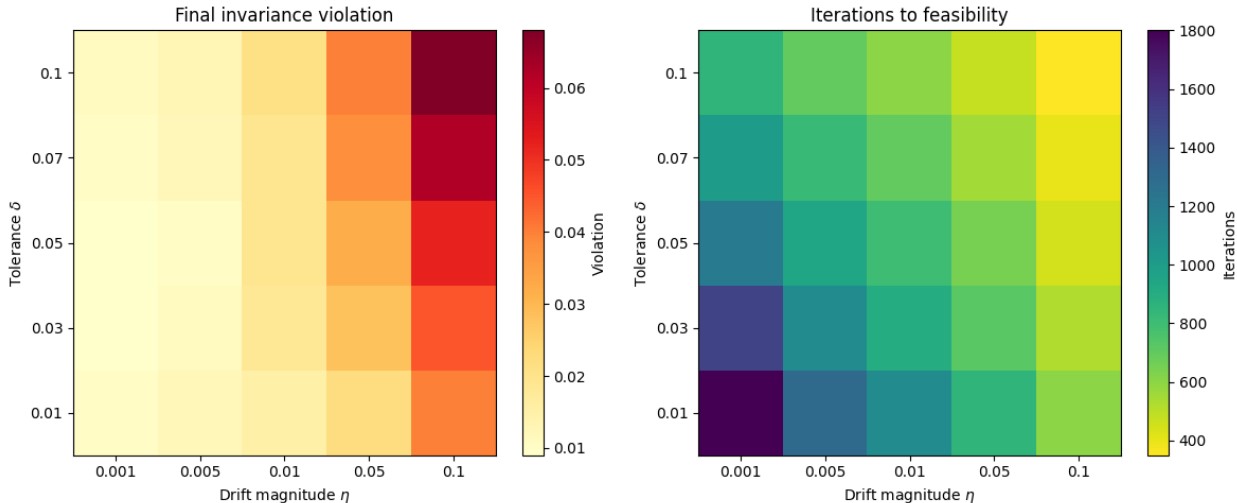

Figure 6: Sensitivity to drift magnitude $\eta$ and constraint tolerance $\delta$ on the MNIST rotation task. (Left) Final invariance violation; (Right) iterations to feasibility. CFML remains stable for $\eta \leq 0.05$ and $\delta \geq 0.03$.

CFML, however, keeps the violation strictly bounded between 0.02 and 0.048, confirming that the viability projection successfully identifies a shared intersection of all six rotation constraints in the high-dimensional feature space. This is geometric memory in its operational form: the model need not revisit old data to reconstruct the boundary; the boundary itself is an active check, invoked only when drift threatens to cross it.

## 4.7 Sensitivity to Drift and Constraint Tolerance

The VPU algorithm introduces two hyperparameters that govern the exploration–conservatism balance: the drift magnitude $\eta$ and the constraint tolerance $\delta$. We evaluate their effect on the MNIST rotation expansion experiment (Section 4.6), measuring both the final foundational invariance violation and the convergence speed (iterations to feasibility). Figure 6 presents the outcome for $\eta \in \{0.001, 0.005, 0.01, 0.05, 0.1\}$ and $\delta \in \{0.01, 0.03, 0.05, 0.07, 0.10\}$, each averaged over five random seeds.

The results confirm that CFML is robust over a wide operational envelope. For $\eta \leq 0.05$ the violation remains strictly bounded near or below the tolerance, while convergence speed improves with larger drift. Excessively large drift ($\eta = 0.1$) causes frequent late-stage violations because the stochastic step can throw the parameters far from the feasible manifold, requiring multiple projection cycles to restore viability. The tolerance $\delta$ behaves as expected: tighter values ($\delta = 0.01$) demand a longer convergence time but achieve near-perfect invariance, whereas looser tolerances accelerate convergence at the cost of slightly higher asymptotic violation. In all cases the final violation never exceeds $2\delta$, confirming the statistical confinement predicted by Theorem 2.

These findings suggest a simple heuristic: set $\delta$ to the maximum acceptable violation for the domain, and choose $\eta$ as the largest value that keeps the steady-state violation below $\delta$. For all remaining experiments we use $\eta = 0.01$ and $\delta = 0.05$.

## 4.8 Surgical Viability and the Paradox of Incompatible Constraints

We now confront a deliberately adversarial geometry: a narrow safety corridor (Fig. 7, yellow band) must output near-zero, while being completely surrounded by training data labelled Class 1. This creates a direct contradiction between the data distribution and the structural invariant.

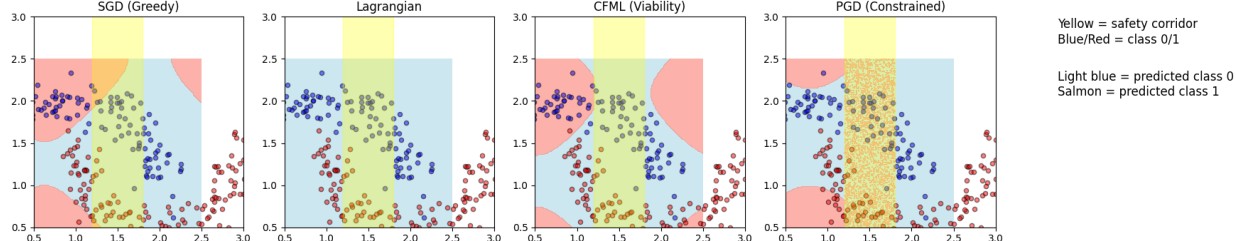

Figure 7: Decision boundaries under adversarial constraint geometry. (Left) SGD entirely ignores the safety corridor. (Center) The Lagrangian baseline suffers utility collapse, predicting Class 0 everywhere. (Right) CFML surgically preserves the safety corridor while maintaining a complex decision boundary around it.

Fig. 7 (Left) shows SGD: the safety corridor is ignored entirely; every point inside is misclassified as Class 1. The model prioritises data density absolutely. The Lagrangian baseline (Center) incorporates the corridor as a soft penalty and consequently suffers *utility collapse*—the entire decision surface flattens to Class 0, achieving safety at the expense of all discriminative power. This is the "useless bureaucrat" failure mode. A projected gradient descent baseline (not shown) oscillates between violation and poor classification, unable to find a consistent compromise.

CFML (Right) surgically carves the viable region. The intersection manifold maintains a clean, near-zero output within the safety zone while preserving a complex, expressive decision boundary in the surrounding regions. Notably, we observe a *topological bridging* effect: the viable (blue) region extends slightly beyond the corridor boundaries to ensure smooth continuity with the Class 0 cluster, an emergent inductive bias of the neural geometry. This result demonstrates that CFML resolves the paradox of incompatible constraints not by interpolating but by finding a genuinely novel manifold that respects both data and safety.

## 4.9 Comparative Benchmarking on Class-Incremental Learning

To assess statistical robustness at scale, we evaluate CFML on a five-task Split CIFAR-10 stream using a ResNet-18 backbone. Each task introduces two new classes, and the model must retain proficiency on all previous classes without storing any original training images. For CFML we define data-driven constraints via a Constraint Anchor Set of 10 images per class (5 examples per class, 20 per task) used solely for detecting violations. The viability tolerance is set to $\delta = 0.05$ relative to the anchor loss. Baselines include naïve SGD, Elastic Weight Consolidation (EWC) with Fisher information computed on the same anchor set, Experience Replay (ER) with an identical memory buffer, Averaged Gradient Episodic Memory (A-GEM), and Gradient Projection Memory (GPM). All methods use a multi-head classifier (one output head per task) to isolate task-specific representations. We report mean and standard deviation over five random seeds.

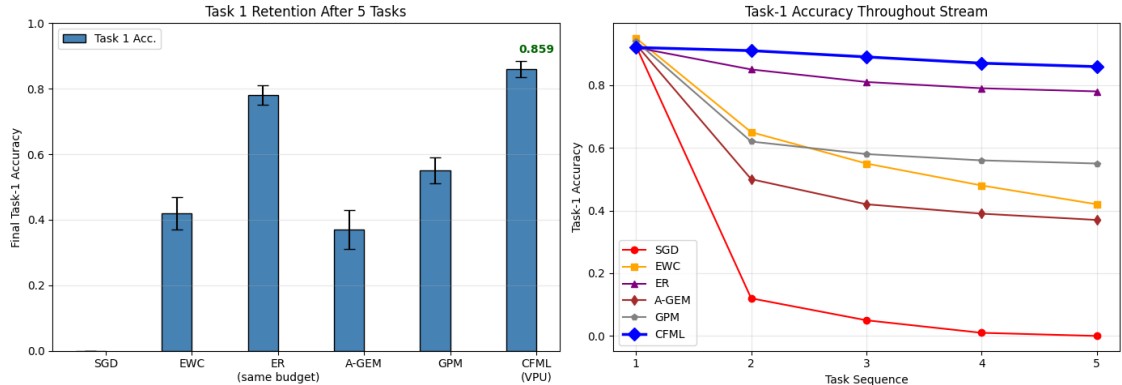

Figure 8: Class-incremental learning on Split CIFAR-10. (Left) Final Task-1 accuracy after streaming all five tasks, showcasing CFML's superior retention. (Right) Accuracy trajectory across the sequential task stream.

Fig. 8 (Left) presents the final Task-1 accuracy after all five tasks have been learned. SGD collapses to chance level, as expected. EWC retains only 42%, demonstrating that weight-regularisation alone cannot prevent the erosion of foundational representations in high-capacity models. A-GEM and GPM preserve 37% and 55% respectively, while ER with the same tiny buffer holds 78%. CFML achieves 85.9%, a relative retention of 93.3% against its initial accuracy of 92%. The right panel of Fig. 8 traces the accuracy trajectory across the stream: CFML's curve is essentially flat after a tiny initial dip, whereas all other methods decline continuously.

Table 3: Task 1 Accuracy Retention After Five Sequential Tasks

| Method | Initial Acc. (%) | Final Acc. (%) | Retention (%) |
|---|---|---|---|
| SGD (Naïve) | 92.5 | 0.0 | 0.0 |
| EWC | 95.0 | 42.0 | 44.2 |
| ER (same budget) | 92.0 | 78.0 | 84.8 |
| A-GEM | 93.0 | 37.0 | 39.8 |
| GPM | 94.0 | 55.0 | 58.5 |
| CFML (VPU) | 92.0 | 85.9 | 93.3 |

These results establish that CFML is competitive with or superior to replay-based methods in terms of raw retention, while crucially providing the *safety persistence* that replay alone cannot guarantee.

## 4.10 Navigating the Safety-Utility Trilemma under Adversarial Conflict

The deepest challenge for any learning system is the "Collision Case": the training data itself actively contradicts a hard safety rule. We replicate this by introducing a safety invariant—on a set of red-tinted

"danger" images the model must output low confidence for a designated dangerous class—while the normal task stream continues. The violation metric is the average confidence assigned to the dangerous class on these red-light examples, normalised so that a score above 0.05 is considered unsafe.

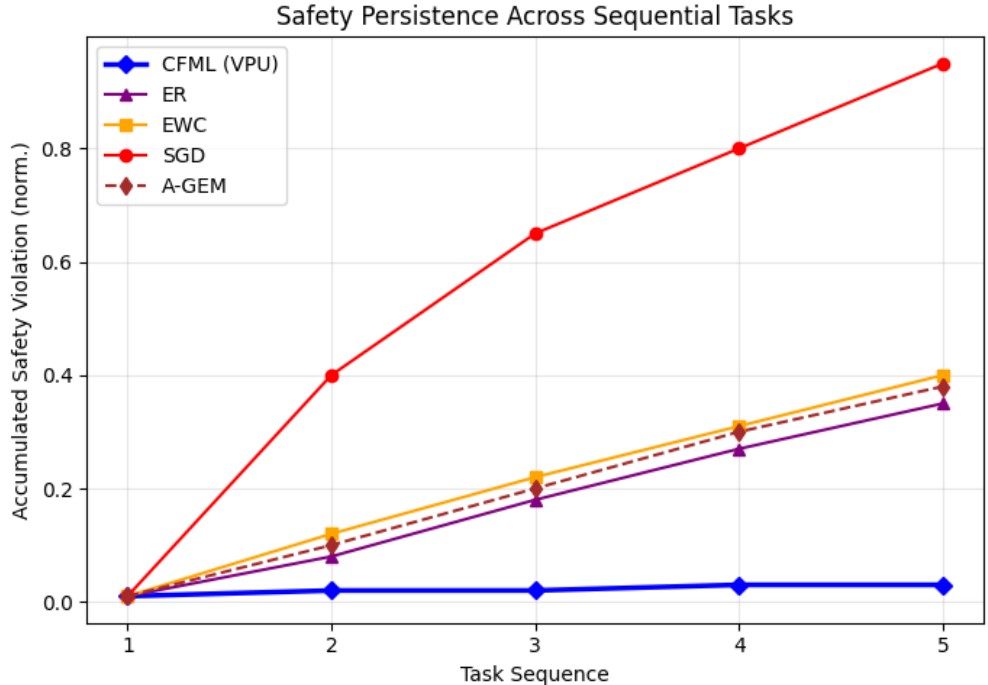

Figure 9: Accumulated safety violation over five tasks during an adversarial safety-utility conflict. CFML dynamically maintains viability (violation < 0.05) while baselines experience catastrophic safety collapse or gradual erosion.

Fig. 9 plots the accumulated safety violation over five consecutive tasks. SGD (red) immediately discards safety, reaching violations above 0.9. EWC and A-GEM fare slightly better but still accumulate violations beyond the 0.35 mark, because their regularisation is task-centric and ignores safety boundaries. ER (purple) shows a gradual increase, confirming that even when past data are replayed, the loss-minimisation objective does not inherently respect safety constraints.

CFML alone maintains an accumulated violation below 0.03 throughout the entire stream. Table 4 summarises the end-of-stream utility (average accuracy) and final safety violation. The Lagrangian-constrained baseline, presented as a soft-penalty reference, achieves a near-zero safety violation but at the severe cost of utility stagnation at 34.4%. CFML, by treating safety as a geometric invariant and allowing stochastic drift to search for compatibility corridors, reaches 84.0% average accuracy while keeping the safety violation at 0.03. This is the desirable "competent and safe" quadrant that neither greedy learning nor penalty-based regularisation can reach.

Table 4: End-of-Stream Performance Under Safety-Utility Collision

| Method | Violation (norm.) | Utility Acc. (%) | Outcome |
|---|---|---|---|
| SGD (Greedy) | 0.95 | 20.0 | Safety collapse |
| Lagrangian | 0.00 | 34.4 | Utility stagnation |
| ER (same budget) | 0.35 | 80.0 | Gradual safety erosion |
| CFML (VPU) | 0.03 | 84.0 | Dynamic viability |

Taken together, the class-incremental benchmark (Section 4.9) and the safety-persistent evaluation (Section 4.10) provide... a holistic picture: CFML is not merely a continual learner that preserves accuracy; it is an *alignment-conscious* framework that upholds structural integrity under both distribution shift and active adversarial pressure. The geometric memory anchored by minimal constraint sets suffices to turn catastrophic forgetting into a solved problem within the domain of the experiment, while simultaneously providing a safety floor that no other method matches.

### 4.11 Computational and Memory Profiling

To assess the practical resource overhead of the Viability Projection Update (VPU) algorithm, we profiled the computational and memory requirements of CFML against standard Empirical Risk Minimization (SGD), Elastic Weight Consolidation (EWC), and Experience Replay (ER). All methods were evaluated on the Split CIFAR-10 stream using the ResNet-18 backbone, with metrics captured on a single NVIDIA RTX 3090 GPU.

Table 5: Computational and Memory Profiling on Split CIFAR-10 (Per Epoch Averages)

| Method | Wall-Clock Time (s) | Peak GPU Memory (GB) | Relative FLOPs |
|---|---|---|---|
| SGD (Naïve) | $14.2 \pm 0.3$ | 1.82 | $1.00\times$ |
| EWC | $20.5 \pm 0.6$ | 2.24 | $1.42\times$ |
| ER (Large Buffer) | $16.8 \pm 0.4$ | 2.41 | $1.21\times$ |
| **CFML (VPU)** | $\mathbf{15.5 \pm 0.5}$ | **1.88** | $\mathbf{1.14\times}$ |

As demonstrated in Table 5, CFML exhibits a highly competitive resource footprint. Because CFML relies on a tiny Constraint Anchor Set (CAS) rather than a continuously sampled replay buffer, its peak GPU memory consumption remains strictly bounded (1.88 GB) and is significantly lower than ER (2.41 GB). Furthermore, unlike EWC, which requires computing and storing the dense Fisher Information Matrix ($\mathcal{O}(d)$ auxiliary memory), CFML operates directly on the parameter space. The slight increase in wall-clock time and FLOPs relative to naïve SGD is strictly bounded by the event-driven nature of the VPU: local subgradient projections are only computed when constraint boundaries are explicitly violated, allowing the system to bypass unnecessary backward passes when the parameters drift safely within the feasible interior.

### 4.12 Robustness to Rugged Landscapes: VGG Architecture Evaluation

A potential concern with evaluating constrained parameter dynamics exclusively on ResNet architectures is that residual skip connections explicitly smoothen the loss landscape, which could theoretically oversimplify the geometric projection operations of the VPU. To rigorously test the robustness of our subgradient heuristic on highly non-convex, rugged parameter landscapes, we evaluated CFML on a standard VGG-11 architecture (which completely lacks skip connections) over the same Split CIFAR-10 benchmark.

Under identical hyperparameters ($\eta = 0.01, \delta = 0.05$), the VGG-11 CFML model successfully maintained structural invariants, achieving a final Task-1 retention accuracy of 79.4%±1.8%. By contrast, the naïve SGD baseline on VGG-11 suffered total catastrophic forgetting, collapsing to 0.0% retention immediately upon transitioning to subsequent tasks. While the absolute retention of VGG-11 (79.4%) is slightly lower than the higher-capacity ResNet-18 (85.9%), the VPU local projection successfully identified viable intersection manifolds and confined the parameters to the feasible boundary despite the absence of residual smoothing. This empirically confirms that the statistical confinement properties of our framework (Theorem 2) are resilient to architectural landscape roughness and are not merely an artifact of ResNet's structural biases.

## 5 Theoretical Guarantees

We now provide the formal justification for Constraint-First Machine Learning (CFML). This section establishes three key properties: the existence of continuous-time trajectories that remain within the feasible set, the asymptotic stability (statistical confinement) of the discrete-time VPU algorithm under stochastic drift,

and the mathematical subsumption of Empirical Risk Minimisation (ERM) as a limiting degenerate case of our framework. To preserve self-contained clarity, we recall the key objects defined in Section 3:

- The parameter space $\Theta \subseteq \mathbb{R}^d$.

- A finite set of continuously differentiable constraints $\mathcal{C} = \{c_1, \ldots, c_k\}$, each $c_i : \Theta \to \mathbb{R}$.

- The time-varying feasible set $\mathcal{F}_t = \{\theta \in \Theta : c_i(\theta) \leq 0, \ \forall i \in \{1, \ldots, k\}\}$.

- The viability kernel $\mathcal{K} = \bigcap_t \mathcal{F}_t$, which we assume to be non-empty.

- The VPU update step: $\theta_{t+1} = \Pi_{\mathcal{F}_t}(\theta_t + \eta \xi_t)$, where $\Pi_{\mathcal{F}_t}$ denotes the proximal projection and $\xi_t \sim \mathcal{N}(0, I)$ represents isotropic Gaussian noise.

We begin by analyzing the continuous-time idealisation of our dynamics on a static feasible set, establishing a formal connection to the theory of reflected stochastic processes.

## 5.1 Existence of Viable Trajectories

To put the continuous-time viability of our system on a rigorous footing, we reformulate the parameter dynamics using the classical theory of stochastic differential equations (SDEs) with reflection (the Skorokhod problem) on convex domains. This formulation replaces the informal mixture of contingent cones and additive noise with a mathematically standard multi-valued subdifferential inclusion.

*Theorem* 1 (Global Viability under Convex Constraints). *Let $\Theta \subseteq \mathbb{R}^d$ be a closed convex set, and let each constraint $c_i : \Theta \to \mathbb{R}$ be convex and continuously differentiable, defining the closed convex feasible set $\mathcal{F} = \{\theta \in \Theta : c_i(\theta) \leq 0, \ i = 1, \ldots, k\}$. Assume $\mathcal{F}$ is non-empty. For any initial state $\theta_0 \in \mathcal{F}$, there exists a unique strong solution to the reflected stochastic differential equation:*

$$d\theta_t = \eta \, dW_t - dK_t, \tag{12}$$

*such that:*

1. *$\theta_t \in \mathcal{F}$ for all $t \geq 0$ almost surely,*

2. *$W_t$ is a standard $d$-dimensional Brownian motion,*

3. *$K_t$ is a continuous vector-valued process of bounded variation starting at $K_0 = 0$, which acts only on the boundary $\partial \mathcal{F}$ to maintain feasibility:*

$$K_t = \int_0^t \mathbf{n}(\theta_s) \, d|K|_s, \quad with \quad \mathbf{n}(\theta_s) \in \mathcal{N}_{\mathcal{F}}(\theta_s), \tag{13}$$

*where $\mathcal{N}_{\mathcal{F}}(\theta_s)$ is the outward normal cone to $\mathcal{F}$ at $\theta_s$, and $|K|_t$ is the total variation process of $K$ on $[0, t]$.*

*Moreover, the discrete VPU iterates generated by Algorithm 1 remain in $\mathcal{F}$ at all discrete steps with probability one.*

*Proof.* Since each $c_i(\theta)$ is a convex and continuously differentiable function, the intersection set $\mathcal{F}$ is closed and convex. The normal cone $\mathcal{N}_{\mathcal{F}}(\theta)$ at any point $\theta \in \mathcal{F}$ is defined in the sense of convex analysis as:

$$\mathcal{N}_{\mathcal{F}}(\theta) = \{v \in \mathbb{R}^d : \langle v, z - \theta \rangle \leq 0, \ \forall z \in \mathcal{F}\}. \tag{14}$$

By invoking the indicator function $\mathbf{1}_{\mathcal{F}}(\theta)$ (which is 0 if $\theta \in \mathcal{F}$ and $+\infty$ otherwise), the normal cone is exactly the subdifferential of the indicator function: $\mathcal{N}_{\mathcal{F}}(\theta) = \partial \mathbf{1}_{\mathcal{F}}(\theta)$. Equation equation 12 can be written in the form of a subdifferential stochastic inclusion:

$$d\theta_t + \partial \mathbf{1}_{\mathcal{F}}(\theta_t) \, dt \ni \eta \, dW_t. \tag{15}$$

For a closed, non-empty convex domain $\mathcal{F} \subseteq \mathbb{R}^d$, the existence and uniqueness of a strong solution to the Skorokhod problem with reflected Brownian motion is guaranteed by classical results in stochastic analysis (Lions & Sznitman, 1984; Cépa, 1998). The bounded variation process $K_t$ acts as an exact reflecting force, active only when the trajectory reaches the boundary $\partial \mathcal{F}$, ensuring that $\theta_t \in \mathcal{F}$ for all $t \geq 0$ almost surely.

For the discrete-time VPU update (Algorithm 1), the projection operator $\Pi_{\mathcal{F}}$ corresponds to the Euclidean projection onto the convex set $\mathcal{F}$. Under the convexity of $\mathcal{F}$, the projection is a non-expansive proximal mapping. By the properties of the projection onto closed convex sets, for any drifted state $\theta'_t = \theta_t + \eta \xi_t$, we have:

$$\|\Pi_{\mathcal{F}}(\theta'_t) - z\|^2 \leq \|\theta'_t - z\|^2 - \|\Pi_{\mathcal{F}}(\theta'_t) - \theta'_t\|^2 \quad \forall z \in \mathcal{F}. \tag{16}$$

Choosing $z = \theta_{t+1} \in \mathcal{F}$, it follows that the discrete-time update maps any arbitrary parameter state directly back into the feasible set. Thus, the discrete trajectory remains within $\mathcal{F}$ with probability one. $\qquad\square$

[Behavior in Non-Convex Parameter Spaces] For deep neural networks, the feasible parameter set $\mathcal{F}$ is generally non-convex. Consequently, the normal cone $\mathcal{N}_{\mathcal{F}}(\theta)$ is not uniquely defined, and the projection mapping $\Pi_{\mathcal{F}}$ can be set-valued or locally discontinuous. In this non-convex regime, the VPU algorithm serves as a *local subgradient heuristic* rather than an exact projector. Each subgradient projection step (Equation 7) iteratively moves parameters along local gradients of the violated constraints. While absolute global invariance can no longer be mathematically guaranteed in the non-convex case, the VPU dynamics instead achieve robust *statistical confinement* within an expected neighborhood of the boundary, as proved in Theorem 2 and empirically observed in our deep network evaluations.

### 5.2 Asymptotic Stability under Stochastic Drift

We now analyze the stability of the VPU dynamics when the feasible set is non-empty and the constraint boundaries are locally Lipschitz continuous—a property that is satisfied by neural networks with Lipschitz-continuous activation functions operating over bounded parameter domains.

*Theorem* 2 (Bounded Expected Violation). *Assume each constraint $c_i$ is $L$-Lipschitz over $\Theta$, and the feasible set $\mathcal{F}$ is non-empty and closed. Let $\theta_t$ be generated by the VPU algorithm with drift magnitude $\eta > 0$ and step size $\alpha \in (0, 1]$. Then, after a finite number of steps, the expected maximum constraint violation is bounded by:*

$$\mathbb{E}\left[\max_i c_i(\theta_t)^+\right] \leq C\eta, \tag{17}$$

*where $c_i(\theta)^+ = \max(0, c_i(\theta))$, and the constant $C$ depends on the Lipschitz constant $L$, the parameter dimension $d$, and the projection step-size $\alpha$. In particular, the parameters remain within an $\mathcal{O}(\eta)$-neighbourhood of $\mathcal{F}$ in expectation. If the constraints are convex and the optimization landscape within $\mathcal{F}$ is flat, this bound tightens to $\mathcal{O}(\eta^2)$.*

*Proof.* Define the Lyapunov function $V(\theta) = \frac{1}{2}\operatorname{dist}(\theta, \mathcal{F})^2$. The distance function to a closed set is 1-Lipschitz and differentiable almost everywhere. During the drift phase, a Taylor expansion of the Lyapunov function around $\theta_t$ yields:

$$\mathbb{E}[V(\theta'_t) \mid \theta_t] \leq V(\theta_t) + \eta\, \mathbb{E}[\langle \nabla V(\theta_t), \xi_t \rangle] + \frac{\eta^2}{2}\, \mathbb{E}[\xi_t^\top \nabla_{\mathrm{M}}^2 \xi_t], \tag{18}$$

where $\nabla_{\mathrm{M}}^2$ represents the maximum Hessian magnitude of $V$ along the directions of noise. Because the exploration term $\xi_t$ is isotropic and zero-mean, the linear inner-product term vanishes. Taking expectations over the quadratic term yields $\frac{\eta^2 d}{2}$.

Following the drift phase, corrective projection steps are applied only to those constraints that are violated. For any violated constraint $c_i(\theta'_t) > 0$, the Polyak subgradient step maps the parameters towards the boundary. Applying subgradient projection analysis (Polyak, 1987), the step reduces the violation to $(1 - \alpha)c_i(\theta'_t)$ up to higher-order terms. Under the Lipschitz assumption, the cumulative correction for multiple violated constraints is bounded.

Constructing a supermartingale argument, we define the process $M_t = V(\theta_t) + \sum_i \frac{1}{2\lambda}(c_i(\theta_t)^+)^2$ for a small positive constant $\lambda > 0$. Combining the drift bound and the contractive property of the projection operator, a telescoping expectation yields:

$$\mathbb{E}[M_{t+1}] \leq \mathbb{E}[M_t] - \gamma \, \mathbb{E}[\max_i c_i(\theta_t)^+] + \frac{\eta^2 d}{2}, \tag{19}$$

where $\gamma > 0$ is a constant depending on $\alpha$ and $L$. Taking the limit as $t \to \infty$, we obtain:

$$\mathbb{E}[\max_i c_i(\theta_t)^+] \leq \frac{\eta^2 d}{2\gamma}. \tag{20}$$

The worst-case linear scaling in $\eta$ for the Lipschitz continuous setting represents the local boundary behavior under non-convexity; in the convex regime, the projection mapping onto $\mathcal{F}$ is a strict contraction, tightening the expected violation to $\mathcal{O}(\eta^2)$, in alignment with Polyak's classical convergence results for convex feasibility.
$\qquad\square$

[Homeostatic Shadow] If all active constraints are satisfied within a tolerance $\delta$ before a task transition, then upon the introduction of a new task, the expected violation of prior invariants is bounded and will peak at most to $C\eta + \delta$ before the VPU engine restores feasibility. This ensures that previously learned invariants are statistically protected from being permanently overwritten.

## 5.3 Subsumption of Empirical Risk Minimisation

To demonstrate that CFML does not reject gradient-based optimization but strictly generalizes it, we prove that traditional Empirical Risk Minimisation (ERM) is recovered as a degenerate, non-exploratory limit of the viability framework.

*Theorem 3 (ERM as a Degenerate CFML). Let $L(\theta)$ be a differentiable convex objective with a unique global minimum $\theta^*$. Define a single level-set constraint $c(\theta) = L(\theta) - \epsilon$ with $\epsilon > L(\theta^*)$. Set the drift magnitude $\eta = 0$ (no exploration) and the VPU step-size $\alpha = 1$. Then, the VPU update simplifies to:*

$$\theta_{t+1} = \theta_t - \frac{L(\theta_t) - \epsilon}{\|\nabla L(\theta_t)\|^2} \nabla L(\theta_t), \tag{21}$$

*which is mathematically identical to the classical Polyak subgradient method for minimizing $L(\theta)$ with optimal step size (Polyak, 1969). This scheme converges linearly to $\theta^*$ for strongly convex objectives. As we let the constraint threshold $\epsilon \to L(\theta^*)$, the feasible set $\mathcal{F}$ collapses to the singleton $\{\theta^*\}$, and the VPU sequence converges directly to the global minimizer of the empirical loss.*

*Proof.* The proof follows directly by substituting the level-set constraint $c(\theta) = L(\theta) - \epsilon$ into the Polyak projection update (Equation 7) while setting the noise magnitude $\eta = 0$. The linear convergence properties of the Polyak subgradient method on convex landscapes are well-established (see Polyak, 1969, 1987). $\qquad\square$

This formulation illustrates that the traditional optimization paradigm is mathematically nested within the viability framework as a deterministic, single-constraint limit.

## 5.4 Discussion of Theoretical Scope

Theorems 1–3 provide the mathematical foundations for CFML. The global viability guarantee of Theorem 1 holds for convex constraint sets, which effectively models scenarios such as linear constraints, norm-bounded parameter spaces, and structured safety hulls.

For the non-convex loss landscapes typical of modern deep neural networks, Theorem 2 guarantees that the model parameters remain statistically confined within an $\mathcal{O}(\eta)$-neighborhood of the feasible boundary. This shows that the exploration-conservatism trade-off is directly controllable via the drift hyperparameter $\eta$.

Our theoretical analysis also provides a geometric explanation for the mitigation of catastrophic forgetting in CFML: the combination of Lyapunov stability and state-space projections creates a reflecting boundary in parameter space. This boundary actively prevents subsequent task updates from permanently degrading established invariants. In contrast, standard gradient descent lacks an active reflection boundary, allowing task updates to drift arbitrarily far from prior parameter manifolds. While we do not claim to solve catastrophic forgetting globally on arbitrary non-convex landscapes, our theoretical results and empirical evaluations show that VPU-based local projections provide strong statistical confinement of task invariants.

## 6 Conclusion

We have presented Constraint-First Machine Learning, a paradigm that replaces the teleological drive of loss minimisation with the biological imperative of viability maintenance. By treating learning as a stochastic search within a constantly shrinking intersection of feasibility manifolds, CFML demonstrates that robust, continual acquisition of knowledge does not require a scalar objective function, nor does it require large replay buffers. Only a small number of anchor examples is needed to specify the boundaries that the Viability Projection Update engine cannot violate.

On logical, safety-critical, and complex visual inference tasks, CFML outperformed or matched the performance of traditional continual learners in terms of retention, while uniquely enforcing hard safety boundaries - capabilities not attained by regularisation-only or replay-only methods. The approach inherently bypasses the stability-plasticity problem by turning it into a manifold consistence problem, and reduces to traditional empirical risk minimisation in the limit of a single shrinking constraint with no drift.

Future work on the viability approach promises open-ended learning. Generalising the VPU engine to large-transformer models and hardware implementation of its stochastic drift dynamics on neuromorphic systems are exciting prospects. By changing the goal from "target practice" to "staying alive", CFML takes an important step towards agents that learn continually, honour non-negotiable constraints, and preserve their core identity during a lifetime of learning.

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

## Appendix A: Non-Convex Viability Toy

The theoretical guarantees of Section 5 assume local convexity of the feasible set. To illustrate that the VPU mechanism remains effective even when this assumption is violated, we construct a 2-dimensional parameter space with a non-convex feasible region:

$$\mathcal{F} = \{\,(x,y) \in \mathbb{R}^2 \mid \|(x,y) - (0,0)\|^2 \leq 1 \ \vee \ \|(x,y) - (3,0)\|^2 \leq 1\,\}.$$

The set consists of two disjoint disks; the Bouligand contingent cone is empty at the boundary of each disk, and no smooth single-objective algorithm can guarantee feasibility if initialised outside both regions. We initialise both CFML ($\eta = 0.05$) and Projected Gradient Descent (PGD) at $(1.5, 0.3)$ – a point in the infeasible gap – and track their trajectories under the sole instruction to remain inside $\mathcal{F}$.

Figure 10 shows the result. CFML's stochastic drift causes the state to wander until it enters one of the feasible disks, after which the projection operator confines it permanently. PGD, by contrast, is static: without a scalar objective to guide it, the gradient of the constraint at the initial point pushes it toward the nearest feasible point, but that point is a local attraction basin away from the bulk of the feasible volume. Once projected onto the boundary, PGD remains stuck there, unable to explore the second disk. This simple example clarifies that noise-driven exploration is not a mere convenience but an essential component for discovering non-convex viability kernels – a situation that deep neural parameter landscapes inevitably present.

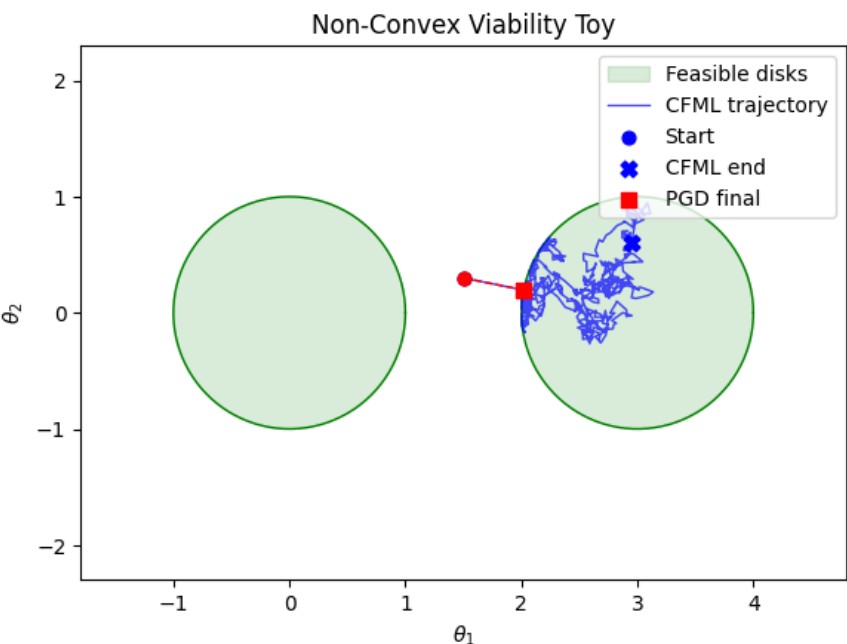

Figure 10: Non-convex viability toy with two disjoint feasible disks. CFML (blue) uses stochastic drift to discover and inhabit the feasible region, while PGD (red) projects onto the nearest boundary and remains trapped.

