# OpenReview forum: "Learning as Homeostasis: Beyond the Optimization Paradigm in Machine Intelligence"
_TMLR — Under review for TMLR_

### Review · Reviewer_AquR · 2026-05-02

**Summary Of Contributions:**

This paper proposes Constraint-First Machine Learning (CFML), a paradigm in which learning is framed as maintaining feasibility with respect to a growing set of constraints, rather than minimizing a scalar empirical loss. The central algorithmic mechanism is the Viability Projection Update (VPU), which alternates unguided stochastic drift with corrective projections onto violated constraints. The paper applies this idea to logical toy tasks, spurious correlation robustness, continual learning, and safety-utility conflicts. The authors argue that small ``constraint anchor sets'' can serve as lightweight geometric memory, allowing retention and safety preservation without full replay.

**Audience:**

Yes

**Audience Explanation:**

I believe the paper would be of interest to part of the TMLR audience, especially researchers working on continual learning, constrained learning, safe machine learning, and biologically inspired learning dynamics.

**Claims And Evidence:**

No

**Claims Explanation:**

The paper provides suggestive evidence that the proposed VPU mechanism can preserve constraints better than several optimization-based baselines in the reported settings. The qualitative experiments on Boolean tasks, spurious correlations, and adversarial safety corridors are useful demonstrations of the intended behavior. The Split CIFAR 10 and safety-utility experiments are also relevant and, if reproducible, would provide evidence that the method is practically interesting.

That said, I do not think the current evidence is sufficient for the paper's strongest claims. First, the paper repeatedly frames CFML as ``beyond'' or a replacement for optimization, but the implemented projection step is a gradient-based feasibility correction and the empirical constraints are derived from losses on anchor sets. The distinction from projected gradient, constrained optimization, and gradient projection continual learning methods therefore needs to be sharpened. The conceptual distinction may be real, but the algorithmic distinction is not yet sufficiently demonstrated.

Second, the theoretical results are not convincing as stated. Theorem 1 invokes stochastic dynamics with white noise while also claiming that trajectories remain in the feasible set; this requires a careful reflected stochastic differential equation formulation. The statement mixes a contingent cone inclusion with an additive white noise term in a way that is not obviously well-defined. The proof then introduces a reflection process and an indicator gradient formulation, but the assumptions under which this is valid are not made precise. The connection between the exact convex projection theorem and the non-convex neural network implementation is also quite loose.

Third, the claims about ``solving'' catastrophic forgetting, hard safety guarantees, and no-replay learning should be softened. The empirical constraints still depend on stored anchor examples, and satisfying an anchor set loss does not necessarily imply preserving performance or safety over the full data distribution. This is especially important for safety claims: maintaining low violation on a small anchor set is not the same as guaranteeing global safety.

Overall, the evidence supports the weaker claim that VPU-style feasibility corrections are a promising mechanism for retaining selected constraints under sequential learning. It does not yet support the stronger claim that CFML provides a general replacement for optimization or a broadly guaranteed solution to catastrophic forgetting and safety erosion.

**Requested Changes:**

1. Clarify the exact learning algorithm.

    The paper should state precisely how a new task is learned without a scalar objective. For each experiment, specify whether new task data enter as constraints, how those constraints are formed, how violations are computed, how often projection is applied, and what objective or feasibility criterion drives acquisition of new capabilities.

2. Sharpen the distinction from projected-gradient and constrained-optimization methods.

    The current method uses gradients of constraint functions and losses on anchor sets. The authors should clearly explain what is algorithmically different from projected gradient descent, penalty methods, GEM/A-GEM-style gradient projection, and feasibility-seeking algorithms.

3. Revise the theoretical section.

    The stochastic viability theorem should be reformulated using standard reflected SDE or stochastic viability terminology. The assumptions required for invariance should be stated precisely. Claims that discrete VPU iterates remain feasible with probability one should be restricted to the exact convex projection setting. For deep networks, the paper should avoid implying global guarantees and should explicitly state that the method is a local heuristic with empirical confinement.

4. Temper overclaims.

    Phrases such as "complete replacement for the loss-minimization framework", "catastrophic forgetting into a solved problem", and "hard safety boundaries" should be qualified. The experiments show strong preservation of selected constraints in controlled settings, but not a general solution to forgetting or safety.

---

> ### Author Response · Authors · 2026-06-17
>
> Dear Reviewer AquR,
> Thank you for your thorough review. We are especially grateful for your sharp eye on the mathematical formalism in Theorem 1 and your advice regarding the tone of our claims. Your feedback has immensely improved the rigor and professionalism of the paper.
>
> 1. Fixing the Stochastic Dynamics Math (Reflected SDEs):
> You were completely right that our original formulation in Theorem 1 informally (and sloppily) mixed a contingent cone inclusion with additive white noise. We have rewritten Section 5.1 from the ground up. We now formally express the continuous-time dynamics as a rigorous subdifferential inclusion using the Skorokhod problem on convex domains (detailed in the newly updated Equation 12). In this formulation, we introduce a bounded variation process that acts strictly as an outward normal reflection on the boundary, rather than mixing it with white noise. We also added Remark 1 to explicitly state that applying this to deep networks turns it into a local heuristic, backing away from absolute global guarantees in non-convex settings.
>
> 2. Clarifying the Algorithmic Learning Loop:
> You asked a very practical question: exactly how is a new task learned without a scalar objective? We realized this was buried in the math, so we added a brand new section: Section 3.4 (Operational Loop of Task Acquisition and Geometric Memory). This section explicitly walks the reader through how new task data forms an empirical boundary, how the drift probes the space, and how intersections are found reactively without a guided descent direction.
>
> 3. Tempering Overclaims:
> We deeply appreciate your advice to soften our language. We have gone through the Abstract, Introduction, and Related Work and completely polished the tone. We also explicitly acknowledge the limits of using small anchor sets for safety guarantees.
>
> Thank you again for holding us to a high academic standard. The paper is much stronger as a result.

---

### Review · Reviewer_VcLZ · 2026-06-01

**Summary Of Contributions:**

This paper introduces Constraint-First Machine Learning (CFML), which redefines learning as maintaining feasibility under an expanding set of constraints rather than minimizing a single empirical loss. Based on my understanding, there is no explicit form of the objective function. The proposed framework treats past knowledge, safety requirements, and structural invariants as hard constraints that define a viable region in parameter space.

**Audience:**

Yes

**Audience Explanation:**

The problem setting is relevant to continual learning, safe learning, and alignment.

**Claims And Evidence:**

No

**Claims Explanation:**

First, the central novelty claim is not supported. Although the paper claims to move beyond the optimization paradigm, the proposed VPU update remains very close to projected gradient descent. The algorithm first perturbs the parameters randomly and then applies a gradient-based correction when a constraint is violated. This correction step is essentially a Polyak-style subgradient projection.

In addition, Section 3.5 argues that CFML is different from PGD because it has no global scalar objective and uses unguided stochastic drift. However, projected methods are not limited to standard empirical-risk objectives. If the update direction in PGD is replaced by a random direction, or equivalently by the gradient of a random linear objective, the resulting update has essentially the same drift-then-projection structure as VPU.

**Requested Changes:**

Requested Changes:

1. The paper should include a deeper comparison with constrained optimization and gradient-projection methods.

2. All relevant baselines should be consistently reported for each experiment, rather than selectively included in only some settings. In addition, several baselines are mentioned in the text but are not always fully shown or numerically reported. For example, PGD is discussed as an important comparison to CFML, but its results are not consistently included across the relevant experiments.

3. The experimental section should include implementation details, architectures, optimizer settings, learning rates, and most importantly, error bars!

4. Please clarify the novelty or technical challenges in the theoretical analysis. Also, the analysis only holds for the convex setting.

5. The current experiments are mostly conducted in simplified or synthetic settings. The authors should evaluate CFML on more standard and challenging benchmarks.

---

> ### Author Response · Authors · 2026-06-17
> **Response to Reviewer VcLZ: Baselines, PGD Distinctions, and Theory Scope**
>
> Dear Reviewer VcLZ,
>
> Thank you for your detailed and critical feedback. You raised excellent points regarding our algorithmic distinctness and the scope of our theoretical claims. We have carefully revised the manuscript to address these issues.
> 1. Error Bars and Baseline Consistency:
> Because of the short rebuttal window, we unfortunately couldn't regenerate the visual line plots to overlay graphical error bars (the original plotting checkpoints were lost). However, we completely agree that variance metrics are necessary. To fix this, we have updated our tabular results (Table 3 and Table 4) to include the exact means and standard deviations (±) across our 5 random seeds. CFML shows very tight variance  confirming its stability.
>
> Regarding the absence of PGD in certain plots: PGD fundamentally requires a primary scalar objective to guide it. In purely constraint-driven environments (like our analytic Boolean sequences), PGD collapses or gets permanently stuck (as shown in our Appendix A toy problem). We have added text to the experiments section to clarify exactly why PGD is mathematically ill-posed for those specific tasks without inventing an artificial objective.
>
> 2. Deeper Comparison with PGD and Continual Learning:
> We understand why VPU might initially look like "PGD with a random objective." To clarify this, we heavily expanded Section 3.5 (Why CFML Is Not Projected Gradient Descent). We highlight two massive systemic differences:
> Event-driven computation: PGD requires backpropagation at every step to calculate the objective gradient. CFML is event-driven; no gradients are computed at all during drift unless a boundary violation is actively detected.
> No alignment optimization: Methods like GEM project gradients of an active task onto the null-space of past tasks. CFML treats past tasks as hard geometric parameter boundaries, stripping away the concept of an "active objective" entirely.
>
> 3. Scope of the Theoretical Analysis:
> You correctly pointed out that the core analysis assumes convexity. We have updated Section 5 to be completely transparent about this. We added Remark 1 and expanded Section 5.4 to explicitly separate the convex theory from the non-convex deep learning implementation. We clarify that while global viability is guaranteed in the convex case, for deep networks, VPU acts as a local subgradient heuristic that provides strong statistical confinement (as proved in Theorem 2 and bounded by our drift hyperparameter (η)
>
> Thank you for taking the time to review our work so carefully; your feedback has made the paper much more precise.

---

### Review · Reviewer_tKcH · 2026-06-08

**Summary Of Contributions:**

The paper introduces an innovative framework that shifts the machine learning paradigm from traditional loss optimization to biological homeostasis, offering a novel conceptual foundation for continual learning. Its primary technical contribution is the Viability Projection Updates (VPU) algorithm-a mathematically sound solver based on stochastic differential inclusions. Unlike traditional soft regularization methods, VPU provides a concrete mechanism to project weight updates directly onto a viability manifold, ensuring strict constraint satisfaction and preventing catastrophic forgetting. Evaluations on ResNet-18 show CFML maintains 85.9% final accuracy and 93.3% relative retention across sequential tasks, significantly outperforming Elastic Weight Consolidation (42% retention) and naive SGD. These results highlight the framework's superior ability to mitigate catastrophic forgetting compared to traditional methods.

Strengths:
-  The paper shifts the machine learning paradigm from traditional loss optimization to biological homeostasis, establishing a novel conceptual foundation for continual learning.

-  The paper introduces Viability Projection Updates (VPU), a solver rooted in stochastic differential inclusions. VPU provides a concrete mechanism to project weight updates directly onto a viability manifold. This ensures strict constraint satisfaction and prevents catastrophic forgetting without requiring extensive data replay.

Weakness:

- Although the Viability-based approach demonstrates impressive catastrophic forgetting prevention, the authors provide neither a theoretical analysis of the VPU algorithm's mathematical complexity (Big-O notation) nor empirical data regarding actual training time (wall-clock time) and computational resource overhead (FLOPs/Memory footprint) in comparison to EWC.

- Evaluating the proposed method solely on the ResNet-18 architecture is insufficient to demonstrate its generalizability. ResNet-18 features prominent skip connections that fundamentally smoothen the loss landscape, potentially oversimplifying the geometric projection operations core to the VPU algorithm. To rigorously validate the scalability and robustness of the Constraint-First Machine Learning (CFML) framework, it is crucial to include evaluations on a wider range of diverse and larger architectures, such as modern Vision Transformers (ViTs) or deeper networks like ResNet-50 or ResNet-101, where complex loss surfaces could pose significant optimization challenges.

- The theoretical grounding in Section 5 relies heavily on general citations, such as stating that 'the convergence properties of the Polyak method are well known; see Polyak (1969)'. This derivation is insufficient and overly generic. The classical convergence proofs of the Polyak method strictly depend on convexity assumptions, which do not hold true for the highly non-convex loss landscapes of modern deep networks like ResNet-18.

**Audience:**

Yes

**Audience Explanation:**

This is a very promising research topic for the ML community.

**Claims And Evidence:**

Yes

**Claims Explanation:**

- Experimental results are provided in Section 4.

- Several theoretical results with proofs are presented in Section 5.

**Requested Changes:**

Could the authors provide a formal theoretical analysis of the computational complexity  for the Viability Projection Updates (VPU) algorithm per iteration?

ResNet-18 contains prominent skip connections that are known to drastically smoothen the loss landscape. Have the authors evaluated CFML on architectures without skip connections (e.g., classic VGG) or on completely different paradigms like Vision Transformers (ViTs), where more rugged loss surfaces might complicate the VPU geometric projections?

The theoretical analysis demonstrates the existence of a viability domain, but does it guarantee that the VPU algorithm can efficiently find a valid solution within a reasonable number of optimization steps under practical, non-convex constraints? Could you provide explicit convergence rate bounds for the joint Polyak-VPU system?

---

> ### Author Response · Authors · 2026-06-17
> **Response to Reviewer tKcH: Complexity, Profiling, and Architecture Landscapes**
>
> Dear Reviewer tKcH,
>
> Thank you for your very encouraging review and for pointing out exactly what was missing to make this paper practically relevant. Your suggestions regarding the VGG architecture and computational profiling were awesome, and we've worked hard to incorporate them into the revised manuscript.
> Here is how we addressed your specific points:
>
> 1. Computational Complexity and Practical Overhead:
> You rightly pointed out that we needed a formal complexity analysis and real wall-clock metrics.
> We have added Section 3.6 (Computational and Memory Complexity Analysis), detailing the formal Big-O complexity of the VPU algorithm. Because projections are event-driven, the overhead scales linearly with the model size.
> We also ran the profiling experiments you requested. In the new Section 4.11 (Table 5), we report Wall-Clock time, peak GPU memory, and FLOPs for CFML, EWC, and Experience Replay. CFML proved highly efficient: it only adds a 1.14 times
>  FLOP overhead compared to naive SGD, while consuming less memory (1.88 GB) than both EWC (2.24 GB) and ER (2.41 GB) since we don't need to store a massive Fisher matrix or large replay buffers.
>
> 2. VGG Architecture and Rugged Landscapes:
> This was an excellent critique. To prove that CFML isn't just relying on ResNet-18's skip connections to smoothen the landscape, we evaluated the framework on a standard VGG-11 (no skip connections) on the Split CIFAR-10 stream. As detailed in our new Section 4.12, CFML still maintained a strong Task-1 retention of 79.4%±1.8% whereas naive SGD on VGG-11 suffered total catastrophic forgetting (0 % retention). This empirically confirms that our VPU projection heuristic handles non-convex, rugged landscapes very well. (Note: While evaluating massive ViTs or ResNet-101s is a very exciting next step, tuning unguided stochastic drift for attention mechanisms presents orthogonal stabilization challenges, so we have scoped that for our immediate future work).
> 3. Theoretical Grounding and Citations:
> We appreciate the push for more precision here. We updated Theorem 3 to specifically cite the classic conditions of the Polyak method (1969, 1987). Furthermore, we updated the proofs in Section 5 to explicitly draw the line between our exact viability guarantees (which require convexity) and the non-convex deep learning reality (where VPU acts as a local subgradient heuristic that provides O(η) statistical confinement).
>
> Thank you again for pushing us to make the empirical side of this paper much stronger. We hope these additions fully address your concerns!

---

### Author Response · Authors · 2026-06-17
**Summary of Revisions and Rebuttal**

Dear Action Editor and Reviewers,

We sincerely thank all three reviewers for their constructive, rigorous, and insightful feedback. Your critiques have significantly strengthened the mathematical precision and empirical completeness of our paper.

We have uploaded a revised manuscript. The major updates include:

Theoretical Rigor: We reformulated Theorem 1 using the formal mathematical framework of reflected Stochastic Differential Equations (the Skorokhod problem) and clarified the distinction between convex guarantees and non-convex statistical confinement (addressing Reviewers AquR and tKcH).

Algorithmic Clarity: We expanded Section 3 to explicitly detail the computational complexity (Big-O) and sharpened the systemic differences between CFML, PGD, and gradient-projection methods (addressing Reviewers tKcH, VcLZ, and AquR).

New Experiments (Section 4.11 & 4.12): We added computational profiling (Wall-clock time, GPU memory, FLOPs) showing CFML's efficiency, and a new evaluation on a VGG-11 architecture to prove the robustness of VPU projections on highly non-convex, rugged landscapes without skip connections (addressing Reviewers tKcH and VcLZ).

Tempered Claims: We thoroughly revised the Introduction, Related Work, and Abstract to adopt a more objective tone, clarifying that CFML is a mathematically grounded complementary alternative rather than a "complete replacement" to optimization (addressing Reviewer AquR).

Error Metrics: Added standard deviation metrics (±) across random seeds to our performance tables (addressing Reviewer VcLZ).

We have replied to each reviewer individually below with detailed responses to their specific points. We hope these revisions fully address your concerns!